# Decay of Free Residual Chlorine in Wells Water of Northern Brazil

Taise Ferreira Vargas [1], Célia Ceolin Baía [1], Tatiana Lemos da Silva Machado [1], Caetano Chang Dórea [2] and Wanderley Rodrigues Bastos [1,*]

1   Environmental Biogeochemistry Laboratory (WCP), Federal University of Rondônia (UNIR), Porto Velho-RO 76801-058, Brazil; taisecontatos@gmail.com (T.F.V.); celiaceolin@gmail.com (C.C.B.); tatianalemoss@gmail.com (T.L.d.S.M.)

2   Department of Civil Engineering, University of Victoria, Victoria, BC V8P 5C2, Canada; caetanodorea@uvic.ca

*   Correspondence: bastoswr@unir.br; Tel.: +55-69-2182-2122

**Abstract:** The concentration of chlorine in water declines as it reacts with various substances, causing decay of the residual free chlorine until its total consumption. In light of the typical characteristics of the water from protected dug wells and tube wells, this study aimed to evaluate the decay kinetics of free chlorine in the water of alternative individual supply (AIS) solutions used in the city of Porto Velho in the Brazilian Amazon region. The free chlorine decay constant in the water was evaluated by "bottle tests," applying a first-order model. According to the results, the type of well and initial chlorine concentration significantly influences the free chlorine decay speed. The water samples from the tubular wells had lower chlorine demand levels, attributed to their better water quality. The simulation of the residual chlorine decay in the different supply sources is an important tool to support safe disinfection processes.

**Keywords:** drinking water quality; disinfection; chlorine kinetics; Brazilian Amazon; free chlorination





## 1. Introduction

Adequate water supply, in quantity and quality, is a fundamental human need and directly affects human health. However, in many regions of Brazil, this supply is substandard relative to national benchmarks. According to data referring to 2018 from the Trata Brazil Institute [1], only 36.7% of the urban population of the city of Porto Velho (state of Rondônia) received treated water from the piped supply system. Thus, more than half of the population must resort to alternative individual supply (AIS) solutions to meet their needs (different from the public water network), mainly in groundwater obtained on shallow dug wells and tubular wells [2]. The exponential increase in the exploitation of underground water resources causes not only a reduction in the recharge of aquifers but also presents a series of challenges regarding the administration and regulation of these resources for Brazil and for cross-border hydro political interactions since these resources transcend international borders [3–5].

In the Brazilian Amazon, protected dug wells (up to 20 m deep) are more vulnerable to contamination and are more susceptible to seasonal variations in yield than deep tube wells (typically greater than 30 m). Contamination can occur in several ways, especially domestic sewage, since only 4.8% of the population of the city of Porto Velho is served by sewage collection systems [1], posing a direct risk of groundwater contamination from decentralized sanitation that serves a large portion of the urban population.

Drinking water generally needs some type of treatment, depending on the raw water characteristics, to ensure the protection of human health. Among the types of water treatment, disinfection with chlorination is the process of choice for the inactivation of pathogenic microorganisms in contexts such as Porto Velho [6,7]. In Brazil, the current regulations on the limits of free residual chlorine in water destined for human consumption

specify levels between 0.2 and 2.0 mg·L$^{-1}$ [8] in the entire length of the water distribution system.

Chlorine is the most used product for water disinfection worldwide. Its cost is relatively low, and it has an active residual effect, meaning its action continues after being applied in the water distribution system [9]. Chlorine reacts with various substances in water and can be consumed through the oxidation of organic and inorganic matter, causing a decline in its concentration. The reaction of chlorine with organic matter, mainly humic substances, leads to the formation of unwanted disinfection byproducts, such as organochlorine compounds (e.g., trihalomethanes), which can pose a risk to human health [10]. The result of these chemical reactions, added to the components of the distribution system, such as the type of material of the pipes and biofilms present on the inner walls of pipes and tanks, among others, contribute to reducing the residual chlorine over time [11] and can be compounded by environmental factors, such as temperature [12].

The amount of chlorine consumed and the speed at which it happens varies from system to system. The mathematical modeling of water supply systems is an important management tool, as it allows simulating the decay of free residual chlorine through its kinetic coefficients "$k_b$" and "$k_w$". The reactions that occur between the chemical species present in the water are called bulk decay ($k_b$), and the reactions that occur at the interface with the walls of pipes or tanks are called wall decay ($k_w$) [13,14].

First-order kinetics models are traditionally applied in studies of chlorine decay due to their simplicity of application since the decay of chlorine depends on the chemical characteristics of the water [15]. The kinetic constant value ($k_b$) for first-order reactions can be estimated from laboratory tests, generically called "bottle tests". This method allows the reactions of chlorine associated with the aqueous medium to be analyzed simply as a function of time, and the graphical representation of the values of ln ($C_t/C_0$) as a function of time allows obtaining a straight line whose slope is the value of $k_b$ [13,14,16].

This study explores chlorine demand and its interactions with the substances present in groundwater sources, considering the local difficulties of a region lacking resources for safely managed water services and with a relatively hot climate, challenging factors the successful implementation chlorination process. Environmental factors, such as water quality and temperature, are important considerations for improving the chlorination by determining the appropriate (i.e., achieving desired free residual chlorine) and acceptable (i.e., regarding taste and odor) initial dosages by users.

The objective of this study was to determine the kinetic constants of chlorine decay in the water of alternative individual supply systems in Porto Velho, Rondônia, as well as to associate the decay of free residual chlorine with the physical–chemical and microbiological characteristics, types of water sources and initial dosage of chlorine.

## 2. Materials and Methods

### 2.1. Study Area

The study was carried out in the urban area of the city of Porto Velho (Figure 1), capital of the state of Rondônia, located in the North region of Brazil (08°45′43″ S, 63°54′14″ W), currently with about 500,000 inhabitants. The study covered ten residences without access to treated public water supply. They all obtained water from wells, five from shallow wells (protected dug wells) and the other five from deep tube wells, all of them equipped with submerged pumps for water extraction and piped water systems. This study was approved by the research ethics committee of the Federal University of Rondônia (UNIR) in May 2019, under registration No. 3.361.927.

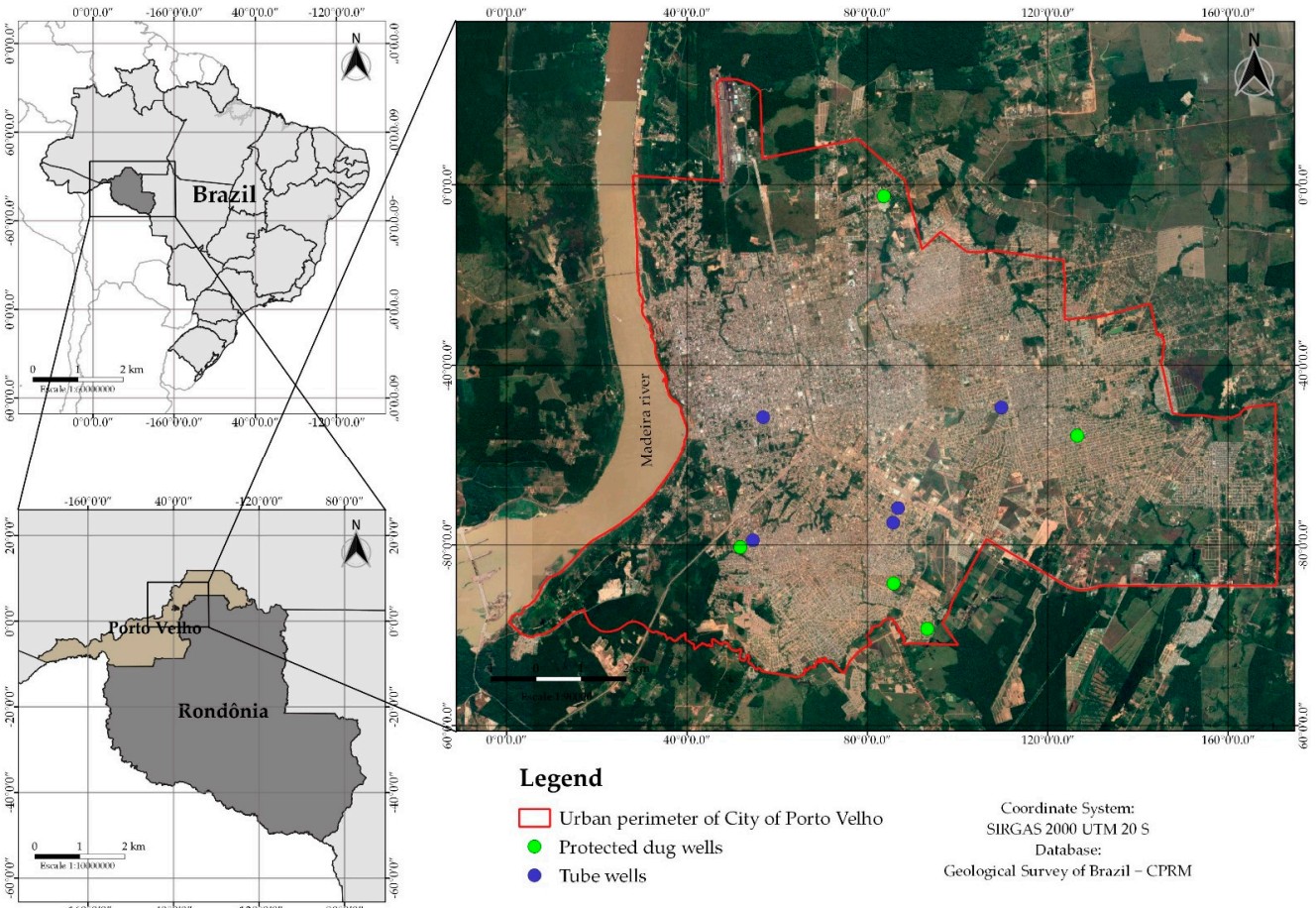

**Figure 1.** Study area showing the sampled wells in Porto Velho.

They all obtained water from wells, five from shallow wells (protected dug wells) and the other five from deep tube wells.

Water samples were collected in January and February 2020. All amber glass bottles used in the experiments were previously washed with neutral detergent (Extran MA 02, Merck S/A, São Paulo, SP, Brazil) and $20.0 \text{ mg} \cdot \text{L}^{-1}$ of a sodium hypochlorite solution (Dinâmica® Química, São Paulo, Brazil) to remove chlorine reactive substances and were subsequently rinsed with deionized water [12]. The water samples were collected in duplicate, the exit of each well, following all the Brazilian guide's recommendations for sample collection and preservation [17]. The bottles were stored in the dark under ice-cooling sent immediately to the Environmental Biogeochemistry Laboratory (WCP) of Federal University of Rondônia (Porto Velho, Rondônia, Brazil) for analysis. The parameters pH, electrical conductivity, temperature and turbidity were measured in loco using a multiparameter probe (Bante 900P, Sonic Supply, Woodbury, CT, United States) and turbidimeter (Akso TU430, São Leopoldo, Rio Grande do Sul, Brazil). The determination of ionic constituents, nitrate and phosphate, was performed by ion chromatography (Metrohm Brazil —model 882 Compact IC plus, Perdizes, São Paulo, Brazil).

Prior to performing the chlorine decay test, a sodium hypochlorite solution (Lafepe, Recife, Pernambuco, Brazil) was prepared by dilution to a free chlorine concentration of $20.0 \text{ mg} \cdot \text{L}^{-1}$ in ultrapure water (Milli-Q, Millipore, Burlington, MA, USA). The determination of the kinetic constants in the liquid mass of water was obtained from the "bottle tests" [14,15,18]. For the chlorine decay tests, chlorine was added to amber bottles containing 500 mL of water samples to obtain final concentrations of 0.2, 0.6 and $1.0 \text{ mg} \cdot \text{L}^{-1}$. As a control, the sodium hypochlorite solution was added in ultrapure water (Milli-Q, Millipore, Burlington, MA, USA) to compare chlorine decay in the absence of reactive

substances with chlorine, following the same procedures. Subsequently, the bottles were placed in an incubator at a constant temperature of 25 °C, protected from light. Then, 10.0 mL aliquots of samples were removed at time intervals of 0 h (immediately), 0.5 h, 1 h, 6 h, 12 h, and 24 h. To determine the free chlorine concentrations by the N, N-diethyl-p-phenylenediamine (DPD) method, using a portable chlorine meter (Akso model 404, São Leopoldo, Rio Grande do Sul, Brazil) according to the method described in the Standard Methods for the Examination of Water and Wastewater [19].

### 2.2. Microbiological Analyses

The counting of heterotrophic bacteria was carried out using R2A Agar (Acumedia, Indaiatuba, São Paulo, Brasil). To count of total coliforms and thermotolerant coliforms was carried out using Chromocult® coliform agar (Merck, Millipore, Burlington, MA, USA) following the manufacturer's recommendations. From the water samples collected, serial dilutions were performed to reduce the number of microorganisms per unit volume. Then 100 mL of each sample at each concentration was filtered through a 0.45 μm sterile cellulose membrane. Each sample was analyzed in duplicate, and filtration of the sterile diluent was performed as a negative control. The cellulose membranes were placed on agar in Petri dishes and incubated at 35 °C ± 1 °C for 48 h ± 2 h in a bacteriological incubator. The colony-forming unit (CFU) counts were multiplied by the dilution factor and expressed in CFU 100 mL$^{-1}$ [16].

### 2.3. Statistical Analyses

Initially, descriptive statistics of the water quality parameters of the wells were calculated (mean, standard deviation, median, minimum and maximum). Data normality was ascertained by the Shapiro–Wilk test, and statistical procedures were carried out accordingly, i.e., a two-tailed test was applied to data without normal distribution. The difference in the chlorine decay rate (k) a the water samples of the test and control treatments was analyzed by the Wilcoxon test. To verify the existence of a difference in the decay rate of chlorine (k) between the sample of water from tube wells and protected dug wells, the Mann–Whitney test was used since the treatments were independent but not normally distributed [20]. Two-way ANOVA was used to verify whether the decay rate of chlorine (k) was influenced by the type of well or the concentration of sodium hypochlorite [20,21]. The tests null hypotheses were no difference in the averages of factor A (protected dug wells); no difference in the averages of factor B (tube wells); and no interaction between factors A and B. One-way ANOVA was performed with repeated data for tube wells and protected dug wells, separately, to test for significant difference between chlorine decay speed at different concentrations of sodium hypochlorite. All analyses were performed using the R software (Free Software Foundation, Boston, MA, USA) [22].

## 3. Results

The water samples tested in this study, both from protected dug wells and tube wells, did not undergo any type of treatment before being distributed to each house's storage tank and later for daily use. Table 1 presents the results of the physical–chemical water quality parameters of the protected dug wells and tube wells.

### 3.1. Microbiological Analysis

The investigation of groups of heterotrophic bacteria, total coliforms and thermotolerant coliforms is shown in Figure 2.

The water in eight of the individual supply systems contained total coliforms and thermotolerant coliforms, except two tube wells (AIS 01 and AIS 09). However, heterotrophic bacteria were present in water from all ten wells, with counts below 500 CFU mL$^{-1}$ in the five tube wells, while in the five protected dug wells, the count exceeded that limit indicated by Brazilian regulations [8] for human consumption.

**Table 1.** Physical–chemical water quality parameters in wells in Porto Velho (RO, Brazil).

| Water Source (AIS) | Well Type | Temperature (°C) | Electrical Conductivity ($\mu$S cm$^{-1}$) | pH | Turbidity (NTU) | Nitrate (mg L$^{-1}$) | Phosphate (mg L$^{-1}$) |
|---|---|---|---|---|---|---|---|
| AIS 1 | Tube well | 27.0 | 108.0 | 5.3 | 0.20 | 25.83 | 0.20 |
| AIS 2 | Protected dug well | 27.4 | 120.0 | 5.1 | 15 | 12.76 | 0.16 |
| AIS 3 | Protected dug well | 27.0 | 80.0 | 5.3 | 2.5 | 14.78 | 0.10 |
| AIS 4 | Tube well | 28.1 | 130.0 | 4.8 | 11 | 19.96 | <LD |
| AIS 5 | Protected dug well | 27.0 | 180.0 | 5.9 | 19 | 33.77 | 0.07 |
| AIS 6 | Protected dug well | 28.5 | 57.4 | 5.9 | 6.7 | 7.72 | <LD |
| AIS 7 | Tube well | 28.0 | 59.2 | 4.9 | 4.1 | 6.4 | 0.08 |
| AIS 8 | Tube well | 27.1 | 49.0 | 4.9 | 1.3 | 9.42 | 0.24 |
| AIS 9 | Tube well | 27.9 | 12.6 | 5.5 | 1.1 | 0.1 | 0.06 |
| AIS 10 | Protected dug well | 27.2 | 140.0 | 5.0 | 33 | 10.35 | 0.06 |
| Descriptive Statistics | Min | 27.0 | 12.6 | 4.8 | 0.20 | 0.1 | 0.13 |
| | Mean | 27.5 | 93.6 | 5.3 | 9.5 | 14.1 | 0.24 |
| | Max | 28.5 | 180.0 | 5.9 | 33.0 | 33.8 | 0.07 |
| | SD | 0.6 | 50.6 | 0.4 | 10.0 | 10.0 | 0.20 |

LD: Detection Limit.

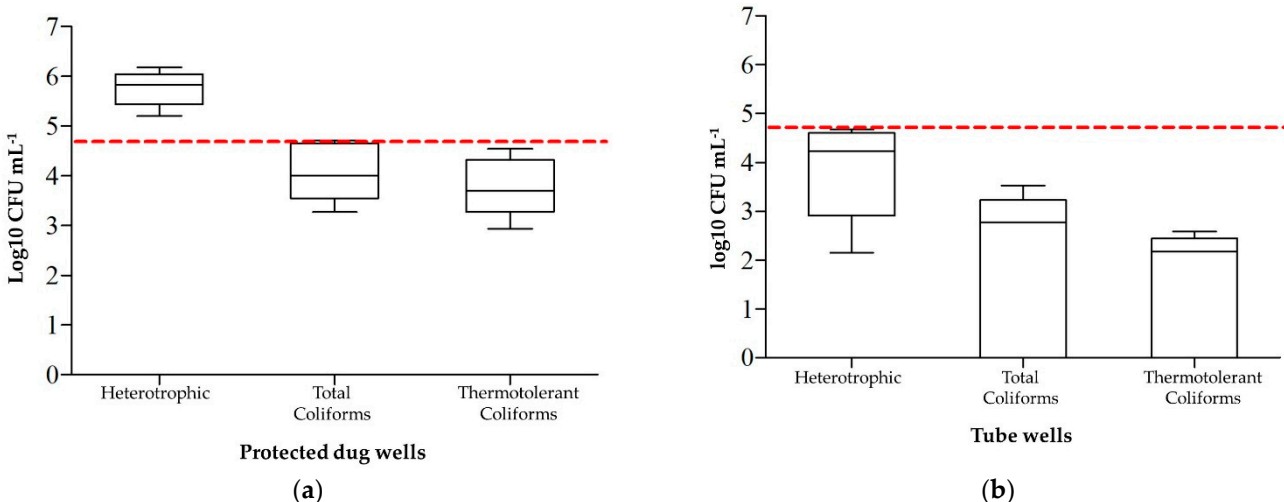

**Figure 2.** Box-and-whisker plots (min/max, lower/upper quartiles, and median) shown into log10 of the counts of heterotrophic bacteria, total coliforms and thermotolerant coliforms in water samples of the protected dug wells ((**a**), n = 5) and tube wells ((**b**), n = 5) in Porto Velho. The dashed line represents the limit heterotrophic bacteria indicated by Brazilian regulations.

### 3.2. Chlorine Decay

The decay kinetics of free residual chlorine was expressed by a first-order model, evaluating the reactions exclusively in the liquid mass of the water through bottle testing, which is an easy method and has been frequently employed in water quality modeling [13,23–25]. To verify the most appropriate initial concentrations to guarantee protective residuals during the storage of water until consumption, a duration of 24 h was established for experiments, considering that the residents collected water from the wells daily. Figure 3a–c show the mean values of ln ($C_t/C_o$) as a function of time of the protected dug wells (n = 5) and tube wells (n = 5), produced from the free chlorine decay test with initial concentrations of 0.2, 0.6 and 1.0 mg·L$^{-1}$ of sodium hypochlorite.

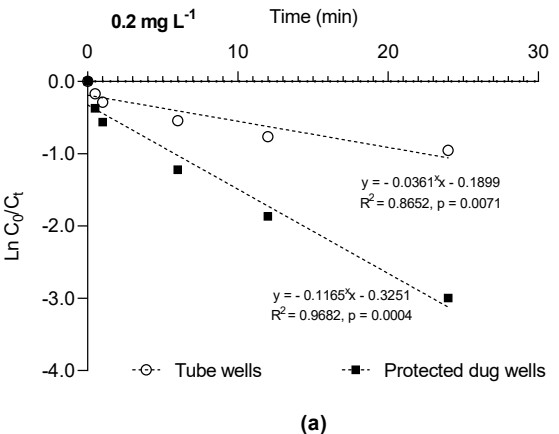

(a)

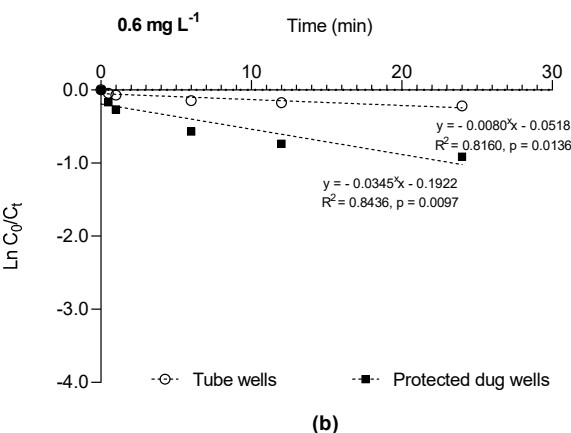

(b)

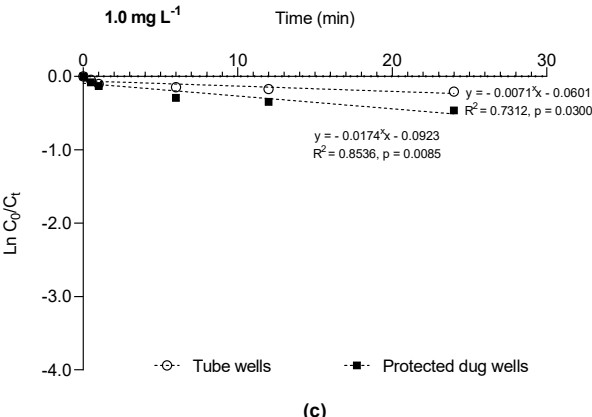

(c)

**Figure 3.** Decay of sodium hypochlorite as a function of time for initial concentrations of 0.2 mg·L$^{-1}$ (**a**), 0.6 mg·L$^{-1}$ (**b**) and 1.0 mg·L$^{-1}$ (**c**).

Considering the first-order model, the well samples (protected dug wells and tube wells) showed an average chlorine decay speed of k = 0.0410 h$^{-1}$ ± 0.0500, significantly higher than the control k = 0.0106 h$^{-1}$ ± 0.0070 (V = 5, *p* < 0.001; Supplementary Materials Table S1), as shown in Figure 4a.

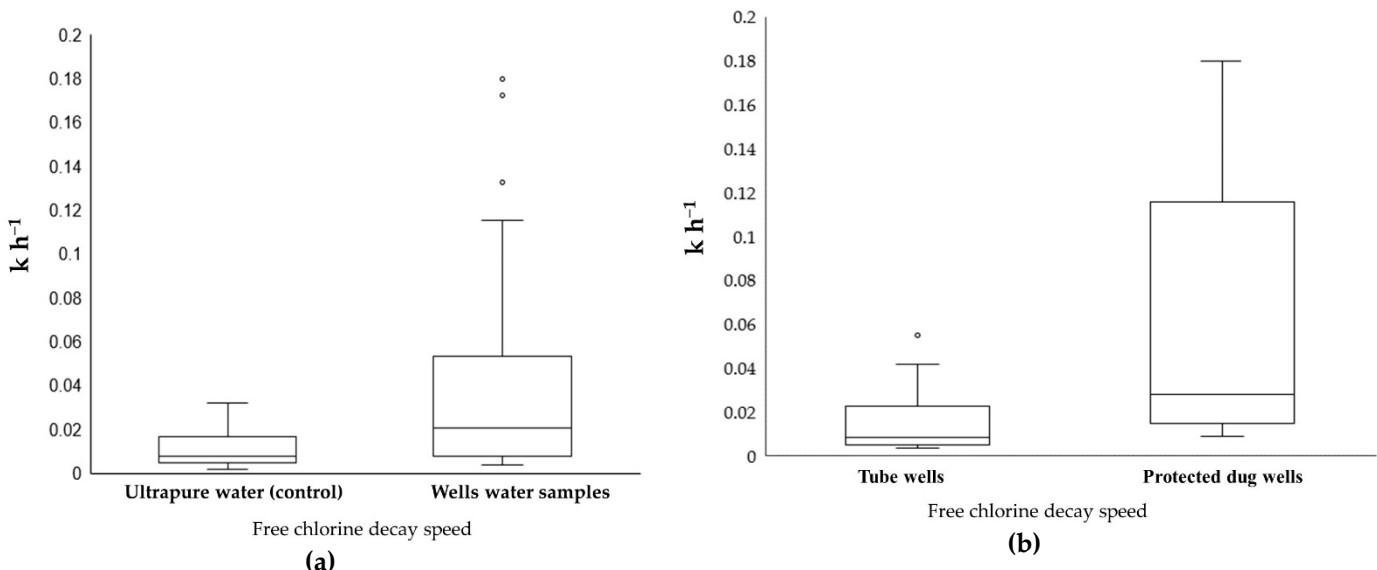

**Figure 4.** Box-and-whisker plots (min/max, lower/upper quartiles, and median) of free chlorine decay speed. (**a**) ultrapure water (control) and well water samples; (**b**) water samples from protected dug wells and tube wells.

Comparing the chlorine decay speed between the water of the protected dug wells vs. tube wells showed average decay speeds of $k = 0.0650$ h$^{-1}$ $\pm$ 0.0610 and $k = 0.0171$ h$^{-1}$ $\pm$ 0.0162, respectively (W: 182, $p = 0.0061$; Supplementary Materials Table S2), as shown in Figure 4b.

The kinetic decay tests of free chlorine in the liquid mass of water ($k_b$) were performed as a function of the residence time of the water until consumption, established in this study as 24 h. Table 2 shows the values of decay coefficients and their respective determination coefficients.

**Table 2.** Decay of the average coefficient of free residual chlorine in the first-order model in the protected dug wells and tube wells of Porto Velho.

| Initial Chlorine Concentration | Tube Wells (n = 5) | | | Protected Dug Wells (n = 5) | | |
|---|---|---|---|---|---|---|
| | **k h$^{-1}$** | **k.day$^{-1}$** | **R$^2$** | **k h$^{-1}$** | **k day$^{-1}$** | **R$^2$** |
| $C_0$ 0.2 mg·L$^{-1}$ | 0.0361 | 0.8664 | 0.8652 | 0.1165 | 2.796 | 0.9682 |
| $C_0$ 0.6 mg·L$^{-1}$ | 0.008 | 0.192 | 0.816 | 0.0345 | 0.828 | 0.8436 |
| $C_0$ 1.0 mg·L$^{-1}$ | 0.0072 | 0.1728 | 0.7312 | 0.0174 | 0.4176 | 0.8536 |

k, kinetic decay coefficients; R$^2$, coefficient of determination.

When analyzing the free chlorine decay constants for 0.2, 0.6 and 1.0 mg·L$^{-1}$ concentrations (Table 2), we observed that the water from the protected dug wells had free residual chlorine after 24 h when the initial concentration was 1.0 mg·L$^{-1}$, and the water in the tube wells when the initial concentration was of 0.6 mg·L$^{-1}$.

A two-way analysis of variance (Figure 5) shows that the type of well and sodium hypochlorite concentration were statistically significant. The concentration of sodium hypochlorite was the most significant variable (F = 22.95, $p < 0.001$). These results indicate that the type of well or sodium hypochlorite concentrations significantly affected the rate of chlorine decay. When evaluating the types of wells separately, one-way ANOVA with repeated measures (Supplementary Materials Table S3) showed an effect of sodium hypochlorite concentration on the chlorine decay rate (F = 33.28, $p < 0.001$) in water samples from tube wells. The comparison between pairs with the Bonferroni test showed that the sodium hypochlorite concentration of 0.2 mg·L$^{-1}$ had the fastest chlorine decay. There were no significant differences in the decay rate at the sodium hypochlorite concentrations of 0.6 mg·L$^{-1}$ and 1.0 mg·L$^{-1}$.

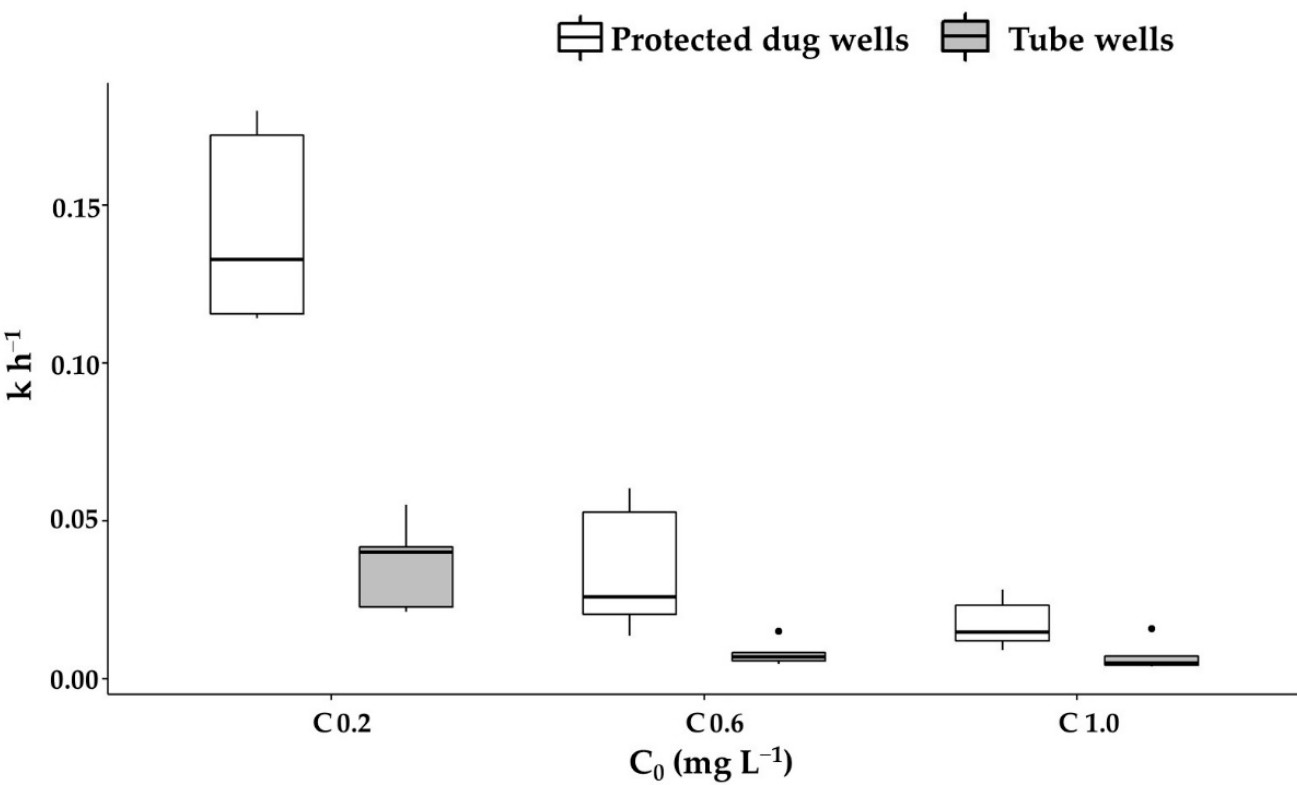

**Figure 5.** Box-and-whisker plots (min/max, lower/upper quartiles, and median) of analysis of variance of the chlorine decay rate in the water samples from protected dug wells and tube wells in Porto Velho.

## 4. Discussion

Water quality parameters have a direct influence on the chlorination process. Regarding temperature, a low variation of this parameter was observed between the groundwater of the protected dug wells and tube wells, with an average temperature of 27.5 °C (Table 1). The chlorine decay rate typically increases with rising temperature [26], but Porto Velho has a low thermal amplitude, and the groundwater is not influenced by atmospheric temperature. In this study, the kinetics of free residual chlorine decay was tested at 25 °C, but we emphasize that rates of free chlorine decay are likely to increase during the hottest hours of the day due to the heating of the water along its path through the tank and distribution pipes.

The presence of contaminants, such as nitrate and phosphate, in groundwater are related to agricultural activities and soil contamination by wastewater (for example, leaking septic systems and sanitary landfills) that end up reaching groundwater by infiltration. According to our data, both nutrients exceeded the limits established by Brazilian legislation, in six wells for nitrate and in seven wells for phosphate [8,27]. The presence of these contaminants can cause health and environmental problems, making the methods of treating these waters for human consumption more complex [28].

It was observed that both water samples from protected dug wells and tube wells showed an acidic pH. The acidity of water favors chemical reactions of the chlorine with certain chemical compounds, which changes the color and flavor of the water and cause stains on clothes and sanitary utensils [29,30], such as waters with high iron concentrations and also raw water containing ammonia and nitrogen compounds that react to form different species of chloramines, also called combined chlorine and are responsible for causing a strong chlorine smell in the water [31,32].

These characteristics of color and smell in the water resulting from the chlorination process can cause the rejection or inadequate application of chlorine, affecting the overall objective of the disinfection process. Although domestic chlorination significantly reduces

the risks of pathogen contamination, the treatment's effectiveness varies according to the chemical composition of the water and the reactions that occur with chlorine. For this reason, technical knowledge is very important for safe and efficient application, in addition to the inclusion of other treatment steps, depending on the characteristics of the raw water.

### 4.1. Microbiological Analysis

The consumption of microbially contaminated water poses a high risk to human health, mainly because it causes many diarrheal diseases, possibly leading to death. Although heterotrophic bacteria are part of the natural microbiota, at high concentrations, they can act as opportunistic pathogens and also hamper the detection of coliforms during laboratory analyses, thus increasing demand for disinfectants [33,34]. Although the tubular wells had heterotrophic bacteria below 500 CFU 100 mL$^{-1}$, we emphasize that these bacteria can adhere inside pipes, forming communities of microorganisms that consolidate themselves through the production of exopolysaccharide matrices (EPS) thus forming biofilms. Biofilms in water distribution systems are a risk to human health since they enhance the resistance of pathogenic microorganisms to disinfection [35].According to the Brazilian Institute of Geography and Statistics (IBGE, 2017). the North of Brazil has the worst rates basic services, such as water supply and/or sanitation [36]. The groundwater contamination by total and thermotolerant coliforms found in this study reflects the effects of the lack of such services in Porto Velho. This is a result of longstanding neglect of the public authorities [37–39], subjecting the population to the risk of developing diseases caused by the consumption of contaminated water. In 2018, 334 people were hospitalized suffering from waterborne diseases in Porto Velho, with nine deaths [40]. Water from dug wells is the first option of many families due to its low cost, but when built without the proper technical criteria and unprotected, for example, near septic tanks, without internal lining and/or with inadequate coverage, these wells are more susceptible to chemical and (mainly) microbiological contamination.

### 4.2. Chlorine Decay

The difference in free chlorine decay speed observed between the well water, and the control samples (Figure 4a) demonstrated that the chemical and biological reactions that occur in the liquid mass of the water have significant interference in the depletion of the free residual chlorine concerning the others analytical interferences used in the methodology. Comparing the chlorine decay rate in water between protected dug wells and tube wells (Figure 4b) indicated that the well type significantly affected the chlorine decay rate, with slower decay in the tubular well water in the three initial chlorine concentrations applied. A study carried in another Brazilian city of similar characteristics, Campo Grande (Mato Grosso do Sul), showed that water of different origins had different chlorine decay profiles [11]. The author observed, using laboratory decay tests, that the first-order decay constant of free chlorine was lower in groundwater (k = 0.0034 h$^{-1}$; Cl$_0$ = 1.0 mg·L$^{-1}$) than in two surface water sources (k = 0.0857 and k = 0.2732 h$^{-1}$; Cl$_0$ = 1.0 mg·L$^{-1}$).

The higher demands for chlorine observed in the protected dug wells suggest a greater presence of organic and inorganic matter in the water in relation to the tube wells, in addition to the microbiological contamination already described. The more reactive substances with chlorine are present, the higher the chlorine doses are needed to maintain a remaining amount that protects water from subsequent microbiological contamination. In practice, the decay rates of free chlorine in the liquid mass found in this study indicate the need for a minimum initial dosage of 0.6 mg·L$^{-1}$ for tubular wells and 1.0 mg·L$^{-1}$ for protected dug wells to maintain a minimum established residual by law for a period of 24 h.

Greater demand for chlorine also is related to greater production of disinfection byproducts (DBPs) during the chlorination process [41]. Exposure to DBPs can occur not only through ingestion of water but also through skin contact and inhalation of water droplets during bathing [42]. As a strategy to minimize chlorine demands, preliminary

treatment procedures should be carried out through coagulation, sedimentation and/or filtration [32].

Chlorine decay was faster when the initial concentration was 0.2 mg·L$^{-1}$ in both well types, and no significant differences were observed in the decay rates at the initial concentrations of 0.6 and 1.0 mg·L$^{-1}$, also in both well types. This suggests that the chemical nature of the contamination responsible for the release of free chlorine is similar.

As soon as chlorine comes into contact with water, it reacts with various substances present at different speeds. Inorganic substances react with chlorine more quickly than organic substances, with speed depending on the complexity and chemical speciation of the compounds [43]. According to Hallam et al. [44] water samples have a fixed number of components that can react with chlorine. Therefore, the fraction of reactive chlorine is higher at lower initial chlorine concentrations [15], as observed in this study. We verified an inverse relationship of initial chlorine concentration and chlorine decay constant in the liquid water mass (Figure 5).

The limitation of this work in determining the kinetics of free residual chlorine-only bulk water. It did not account for other decay factors that could occur in the distribution network, such as chlorine demand from biofilms, the type of pipe materials and the temperature fluctuations and storage times that occur in reservoirs. For these reasons, the values of the initial dosages tend to be higher than the values of 0.6 mg·L$^{-1}$ for the tubular wells and 1.0 mg·L$^{-1}$ for protected dug wells established in this study, for a minimum residual for the period of 24 h, which can result in an aversion of chlorinated water by users, due to taste or smell. According to studies by Crider et al. (2018) [45], chlorine water acceptability limits were 1.16 to 1.26 mg·L$^{-1}$ among low-income residents in Dhaka. However, taste and odor acceptability thresholds are context-specific.

Currently, it is possible to find chlorine of different types in the market, which can be used in the process of disinfecting water from individual alternative solutions (AIS) (diluted sodium hypochlorite solution, NaDCC tablets and granulated calcium hypochlorite) [46–48]. However, it should be noted that the acquisition of these disinfectants implies an increase in household costs. An alternative to performing the water disinfection process of protected dug wells is the use of simplified diffuser chlorinators, as recommended by the Ministry of Health of Brazil [49].

Chlorination is an efficient process to inactivate microorganisms and has been shown to significantly reduce risks to human health. However, it is a complex process that involves many technical requirements to guarantee totally safe water. Hence, there is a need for the public authorities to monitor the quality of groundwater and provide technical assistance and advice to households that obtain water from wells.

## 5. Conclusions

Our results showed that the type of well and the initial chlorine concentration significantly affected the decay rate of free chlorine. The water from the tubular wells presented the lowest demand for free chlorine, attributed to its better initial quality.

We also observed that the values of the initial dosages to guarantee a minimum residual for a period of 24 h were 0.6 mg·L$^{-1}$ for tubular wells and 1.0 mg·L$^{-1}$ for protected dug wells, making it challenging to balance the minimum effective chlorine concentrations and the potentially adverse effects of chlorination on human health.

Our data contribute to the implementation of domestic drinking water chlorination programs and to the development of actions by water treatment managers and operators since alternative solutions for individual supply may be identical to those used in public supply systems.

We believe that the simulation of the decay of residual chlorine in the water from different sources is an important tool to support safe disinfection.

**Supplementary Materials:** The following are available online at https://www.mdpi.com/article/10.3390/w13070992/s1. Table S1: Descriptive statistics of the chlorine decay constants (k) in ultrapure water sample (control) and water from wells located in Porto Velho. Table S2: Descriptive statistics of

the chlorine decay constants (k) in water samples from tubular and excavated wells located in Porto Velho. Table S3: Results of the analysis of variance of data on chlorine decay speed in water samples from tubular and excavated wells located in Porto Velho.

**Author Contributions:** Conceptualization, T.F.V., C.C.D. and W.R.B.; methodology, T.F.V. and C.C.B.; software, T.F.V., T.L.d.S.M. and C.C.B.; validation, T.F.V. and C.C.B.; formal analysis, T.F.V. and T.L.d.S.M.; investigation, T.F.V.; resources, T.F.V. and W.R.B.; data curation, T.F.V.; writing—original draft preparation, T.F.V.; writing—review and editing, T.F.V., C.C.D. and W.R.B.; visualization, T.F.V., T.L.d.S.M. and C.C.B.; supervision, W.R.B.; project administration, W.R.B.; funding acquisition, W.R.B. All authors have read and agreed to the published version of the manuscript.

**Funding:** This study was funded by the Brazilian National Research Council (CNPq, Grant no. 301912/2017-3).

**Institutional Review Board Statement:** The study was conducted according to the guidelines of the National Commission on Ethics in Research - CONEP, and approved by the Research Ethics Committee - CEP of FEDERAL UNIVERSITY OF RONDÔNIA (UNIR) (protocol code 3.361.927 and date of approval 31/05/2019).

**Informed Consent Statement:** Informed consent was obtained from all subjects involved in the study.

**Acknowledgments:** We gratefully acknowledge the staff of the Wolfgang C. Pfeiffer Environmental Biogeochemistry Laboratory of the Federal University of Rondônia.

**Conflicts of Interest:** The authors declare no conflict of interest.

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
