# Peer review of "Decay of Free Residual Chlorine in Wells Water of Northern Brazil"

_water, doi:10.3390/w13070992_

Round 1

Reviewer 1 Report

This paper addresses the problem of determining the kinetic constants of chlorine decay in the water of alternative individual supply systems and the relationship of free residual chlorine decay to the types of water sources, initial chlorine dose, physico-chemical and microbiological properties of the intake and disinfected water. The authors showed that well type significantly affected the rate of chlorine disappearance, with slower disappearance in tube well water at the three initial chlorine concentrations used.
Their results indicate the higher chlorine demand observed in protected dug wells due to the higher presence of organic and inorganic matter in the water compared to tube wells.

According to the results, chlorination is an efficient process for inactivating microorganisms and significantly reduces the risk to human health.
The study was performed correctly, the analysis of the results was carried out properly, which made it possible to formulate correct conclusions. 

I propose to accept the article as presented.

Author Response

We are grateful for the reviewer's considerations.

Reviewer 2 Report

I have read with interest this paper on decay of free residual chlorine in wells water of Northern Brazil. The paper provides a clear research design description in the introduction, and it does operationalise it convincegnly in the methodology and in the discussion sections. My suggestions are:

  • introduction: clearly state the relevant literature you are contributing to, why this study is important, and the gap you are filling
  • in the first intro paragraph, i would suggest to contextualise water issues in Brazil more broadly, therefore discussing how the issue of water supply is particularly important in regions that are at the political border of Brazil, such as in the case of Porto Velho and in other regions of Brazil. See in particular the following articles, which may prove to be useful to contextualise water issues in Brazil of transboundary nature: da Silva, L. P. B. (2019). Production of scale in regional hydropolitics: an analysis of La Plata River Basin and the Guarani Aquifer System in South America. Geoforum99, 42-53; Cassuto, D. N., & Sampaio, R. S. (2011). Keeping it legal: transboundary management challenges facing Brazil and the Guarani. Water international36(5), 661-670; Hussein, H. (2018). The Guarani Aquifer System, highly present but not high profile: A hydropolitical analysis of transboundary groundwater governance. Environmental Science & Policy83, 54-62 ; Brzezinski, M., & Navarro, L. (2010, December). Regulating transboundary groundwater: big challenges for Brazil. In Proceedings of the ISARM2010 international conference “Transboundary aquifers: challenges and new directions (pp. 6-8).

Author Response

Reviewer #2 (marked in red):

I have read with interest this paper on decay of free residual chlorine in wells water of Northern Brazil. The paper provides a clear research design description in the introduction, and it does operationalise it convincegnly in the methodology and in the discussion sections. My suggestions are:

introduction: clearly state the relevant literature you are contributing to, why this study is important, and the gap you are filling

Response & Action: Adjusted. This study explores chlorine demand and its interactions with the substances present in groundwater sources, considering the local difficulties of a region lacking resources for safely managed water services and with a relatively hot climate, challenging factors the successful implementation chlorination process. Environmental factors such as water quality and temperature are important considerations when seeking to improve the chlorination by determining the appropriate (i.e. achieving desired free residual chlorine) and acceptable (i.e. regarding taste and odor) initial dosages by users.

in the first intro paragraph, i would suggest to contextualise water issues in Brazil more broadly, therefore discussing how the issue of water supply is particularly important in regions that are at the political border of Brazil, such as in the case of Porto Velho and in other regions of Brazil. See in particular the following articles, which may prove to be useful to contextualise water issues in Brazil of transboundary nature: da Silva, L. P. B. (2019). Production of scale in regional hydropolitics: an analysis of La Plata River Basin and the Guarani Aquifer System in South America. Geoforum99, 42-53; Cassuto, D. N., & Sampaio, R. S. (2011). Keeping it legal: transboundary management challenges facing Brazil and the Guarani. Water international36(5), 661-670; Hussein, H. (2018). The Guarani Aquifer System, highly present but not high profile: A hydropolitical analysis of transboundary groundwater governance. Environmental Science & Policy83, 54-62 ; Brzezinski, M., & Navarro, L. (2010, December). Regulating transboundary groundwater: big challenges for Brazil. In Proceedings of the ISARM2010 international conference “Transboundary aquifers: challenges and new directions (pp. 6-8).

Response & Action: Adjusted. The exponential increase in the exploitation of underground water resources causes not only a reduction in the recharge of aquifers, but also presents a series of challenges re-garding the administration and regulation of these resources for Brazil and for cross-border hydro political interactions, since these resources transcend international borders[3–5].

Reviewer 3 Report

While the research design and presentation of the results is adequate, the authors have not sufficiently described the need for this research, or the gap in the literature it is trying to address. Chlorine kinetics have long been tested in operational settings, and this paper does not add anything new to the research on this topic. Additionally, the sample size it quite small (n=10) and the importance of the findings has not been made very evident in the manuscript.

I suggest the authors consider presenting this research as a short communication or research brief, in lieu of a full research paper.

Some additional comments:

1. In the introduction, chlorine taste is mentioned as an important factor. Some interesting research on this has been done by Crider, et al.

2. Throughout the manuscript, it is unclear the types of systems that were sampled from. It seems that the samples were taken directly from wells or tube wells, yet the introduction only refers to piped systems. Are these small household mechanized systems? Or are the households storing water in containers in their homes? This seems important to understand the practical application of this research.

3. Some references or discussion to the practice of household water treatment through chlorination (with NaDCC tablets or dilute sodium hypochlorite solution) would be appropriate in both the introduction and the discussion. There has been a lot of prior research in this area in terms of chlorine dosing recommendations, disinfection byproduct formation, and effectiveness of disinfection.

A few minor comments:

Line 16 – This should be “…lower chlorine demand rates” (not “free chlorine demand rates”)

Line 86 –“Without accesses” should be “without access”

Lines 119 – 129 – Please explain when samples were tested in terms of the timing when the FCR was tested. It seems like it only at the initial time of the sampling.

Figure 2: I suggest using the same range on the y-axis scale for both plots.

Author Response

Reviewer #3 (marked in brown):

While the research design and presentation of the results is adequate, the authors have not sufficiently described the need for this research, or the gap in the literature it is trying to address. Chlorine kinetics have long been tested in operational settings, and this paper does not add anything new to the research on this topic. Additionally, the sample size it quite small (n=10) and the importance of the findings has not been made very evident in the manuscript.

Response & Action: Adjusted in the introduction as also requested by the reviewer#2.

I suggest the authors consider presenting this research as a short communication or research brief, in lieu of a full research paper.

Response: Grateful for guidance and we are evaluating the possibility.

Some additional comments:

  1. In the introduction, chlorine taste is mentioned as an important factor. Some interesting research on this has been done by Crider, et al.

Response & Action: We appreciate the suggestion and consider it in the text with the insertion of Crider et al. 2018.

  1. Throughout the manuscript, it is unclear the types of systems that were sampled from. It seems that the samples were taken directly from wells or tube wells, yet the introduction only refers to piped systems. Are these small household mechanized systems? Or are the households storing water in containers in their homes? This seems important to understand the practical application of this research.

Response & Action: Adjusted in the introduction and materials and methods. The samples were collected directly from the wells, after being pumped with submerged pumps, but before going to the hydraulic distribution network and water tanks of the homes.

  1. Some references or discussion to the practice of household water treatment through chlorination (with NaDCC tablets or dilute sodium hypochlorite solution) would be appropriate in both the introduction and the discussion. There has been a lot of prior research in this area in terms of chlorine dosing recommendations, disinfection byproduct formation, and effectiveness of disinfection.

Response & Action: Adjusted in the discussion. Currently it is possible to find chlorine of different types of in the market, which can be used in the process of disinfecting water from individual alternative solutions (AIS) (diluted sodium hypochlorite solution, NaDCC tablets and granulated calcium hypo-chlorite) [45–47]. But it should be noted that the acquisition of these disinfectants implies an increase in household costs. An alternative to perform the water disinfection process of protected dug wells, is the use of a simplified diffuser chlorinators, as recommended by the Ministry of Health of Brazil [48].

 A few minor comments:

Line 16 – This should be “…lower chlorine demand rates” (not “free chlorine demand rates”)

Response & Action: Adjusted.

Line 86 –“Without accesses” should be “without access”

Response & Action: Adjusted.

Lines 119 – 129 – Please explain when samples were tested in terms of the timing when the FCR was tested. It seems like it only at the initial time of the sampling.

Response & Action: Adjusted. The water samples were collected in duplicate, the exit of each well, following all the recommendations of the Brazilian guide to sample collection and preservation [17].

Figure 2: I suggest using the same range on the y-axis scale for both plots.

Response & Action: Adjusted.

Reviewer 4 Report

The manuscript deals with an interesting topic. important all over the world. The research carried out is interesting. it is a pity that the autors took samples only two times, it would be interestin to see how the situation changes in other months - monitoring study at last 6 months.

 It is a pity that  the autors did not take into account the content of nitrates and phosphates in the water from 20-meter wells, with as they claim, is exposed to anthropogenic  polution.

 There were no references to the results of microbiological tests or the quality of water in the inlets.

The conclusions do not stste the purpose of the work. This needs to be completed.

 The manuscript should be corrected.

Author Response

Reviewer #4 (marked in green):

The manuscript deals with an interesting topic. important all over the world. The research carried out is interesting. it is a pity that the autors took samples only two times, it would be interestin to see how the situation changes in other months - monitoring study at last 6 months.

 It is a pity that  the autors did not take into account the content of nitrates and phosphates in the water from 20-meter wells, with as they claim, is exposed to anthropogenic  polution.

Response & Action: We entered nitrate and phosphate data in the table 1.

There were no references to the results of microbiological tests or the quality of water in the inlets.

Response & Action:

The conclusions do not stste the purpose of the work. This needs to be completed.

Response & Action: We adjust the completion as requested.

The manuscript should be corrected.

Response & Action: the manuscript has been corrected.

Round 2

Reviewer 2 Report

Looks good now 

Reviewer 3 Report

comments addressed